# Invasive Validation of Antares, a New Algorithm to Calculate Central Blood Pressure from Oscillometric Upper Arm Pulse Waves

**DOI:** 10.3390/jcm8071073

**Published:** 2019-07-22

**Authors:** Marcus Dörr, Stefan Richter, Siegfried Eckert, Marc-Alexander Ohlow, Fabian Hammer, Astrid Hummel, Vivien Dornberger, Elisabeth Genzel, Johannes Baulmann

**Affiliations:** 1Department of Internal Medicine B, University Medicine Greifswald, Ferdinand-Sauerbruch-Straße, D-17475 Greifswald, Germany; 2German Centre for Cardiovascular Research (DZHK), Partner Site Greifswald, D-17475 Greifswald, Germany; 3Zentralklinik Bad Berka GmbH, Herzzentrum, Department of Cardiology, Robert-Koch-Allee 9, D-99437 Bad Berka, Germany; 4Klinik für Allgemeine und Interventionelle Kardiologie/Angiologie, Universitätsklinik der Ruhr-Universität Bochum, Georgstraße 11, D-32545 Bad Oeynhausen, Germany; 5EPC—European Prevention Center, Luise-Rainer-Straße 6-10, D-40235 Düsseldorf, Germany; 6Department of Medical Psychology and Psychotherapy, Medical University of Graz, Auenbruggerplatz 2/8, A-8036 Graz, Austria

**Keywords:** validation, invasive, central blood pressure, pulse wave analysis, antares

## Abstract

***Background:*** Antares is an algorithm for pulse wave analysis (PWA) by oscillometric blood pressure (BP) monitors in order to estimate central (aortic) blood pressure (cBP). Antares aims to enable brachial cuff-based BP monitors to be type II-devices, determining absolute cBP values independently of potential peripheral BP inaccuracies. The present study is an invasive validation of the Antares algorithm in the custo screen 400. ***Methods:*** We followed entirely the 2017 ARTERY protocol for validation of non-invasive cBP devices, the 2013 American National Standards Institute, Inc./Association for the Advancement of Medical Instrumentation/International Organization for Standardization (ANSI/AAMI/ISO) 81060-2 and 2018 AAMI/European Society of Hypertension (ESH)/ISO validation standard protocols. In total, 191 patients undergoing cardiac catheterization were included, of which 145 patients entered analysis. Invasive cBP recordings were compared to simultaneous non-invasive cBP estimations using the Antares algorithm, integrated into an oscillometric BP monitor. ***Results:*** Mean difference between invasive and non-invasively estimated systolic cBP was 0.71 mmHg with standard deviation of 5.95 mmHg, fulfilling highest validation criteria. ***Conclusion:*** Antares is the first algorithm for estimation of cBP that entirely fulfills the 2017 ARTERY and AAMI/ESH/ISO validation protocols. The Antares algorithm turns the custo screen 400 BP monitor into a type II-device. Integration of Antares into commercially available BP monitors could make it possible to measure PWA parameters in virtually every practice in future.

## 1. Introduction

Central blood pressure (cBP) is not the same as peripheral blood pressure (pBP) [1]. Basically, in a young, healthy arterial system cBP is low and pBP much higher [1]. When an individual is getting older, cBP rises and may reach the same level or probably even higher cBP levels than pBP [2]. Several studies have shown that cBP is more strongly related to hypertension-related organ damage and outcome than pBP [3,4,5]. According to the current European Society of Hypertension/European Society of Cardiology (ESH/ESC) hypertension guidelines, measurement of cBP has practical consequences for the treatment of young patients with isolated systolic hypertension [6]. For these individuals, no medication is necessary if cBP is low. Thus, a potentially huge overtreatment can be avoided. A *conditio sine qua non* should be that cBP is estimated accurately.

For the first time, in 2017 a worldwide consensus paper under the leadership of the ARTERY society was published as a task force consensus statement on protocol standardization for the validation of non-invasive central pressure devices [7]. A main issue was that the reference standard against which the device accuracy of central BP estimations is gauged should be intra-arterial (invasive) catheter in the ascending aorta because “currently there are no non-invasive alternatives”. At the same time, procedures were defined for (a) the proper performance of the invasive pressure recordings and the technical requirements, (b) non-invasive central BP device measurement standards, (c) sample characteristics, and (d) statistical requirements.

The existing devices were categorized as type I or type II devices. Type I means that an estimate of central BP relative to the measured brachial BP is given in order to concentrate on a relatively accurate pressure difference between central and peripheral sites. Type II means that these devices estimate the intra-arterial central BP, what could be understood as the “true” central BP, despite the known inaccuracy of peripheral BP measurements compared to invasive measurements [8].

The ANSI/AAMI/ISO 81060-2:2013 (American National Standards Institute, Inc. and Association for the Advancement of Medical Instrumentation) was developed by the International Organization for Standardization (ISO)/TC 121/SC 3, the International Electrotechnical Commission (IEC)/SC 62D Joint Working Group (JWG) 7 on non-invasive BP monitoring equipment and the AAMI Sphygmomanometer Committee. The objective of the standard is to provide minimum labeling, performance, and safety requirements for the clinical validation of medical electrical equipment used for estimation of the arterial blood pressure by utilizing a cuff. Most, but not all, of the ANSI/AAMI/ISO 81060-2:2013 recommendations are included in the ARTERY Society task force consensus statement on protocol standardization for validation of non-invasive central BP devices [9]. 2018 a statement was published that presents the key aspects of a validation procedure, which were agreed by the AAMI, ESH and ISO representatives as the basis for a single universal validation protocol [10]. Because in the 2018 statements there are no rules for invasive reference measurements, we did not stop referring to the 2013 ANSI/AAMI/ISO protocol with integrated criteria for hemodynamic stability of invasive measurements and finally followed both protocols.

Antares is an algorithm for calculating central blood pressure values, which can be integrated into an oscillometric device at the upper arm. Antares aims to enable an oscillometric BP monitor to act as a type II-device, thereby providing “true” invasive central BP values. Antares was developed by Redwave Medical GmbH, Jena, Germany.

The study aims is to invasively validate the central blood pressure calculation of the Antares algorithm according to the ARTERY validation protocol as well as the 2013 ANSI/AAMI/ISO 81060-2 and the 2018 AAMI/ESH/ISO standard using an oscillometric BP monitor at the upper arm.

## 2. Material and Methods

We followed entirely the 2017 ARTERY society task force consensus statement on protocol standardization for the validation of non-invasive central blood pressure devices, the 2013 ANSI/AAMI/ISO 81060-2 and the 2018 AAMI/ESH/ISO standard.

Table 1: Patient characteristics. Blood pressure (BP) range (systolic, diastolic and mean peripheral and invasive blood pressures), body mass index (BMI), systolic blood pressure (SBP), mean arterial pressure (MAP), diastolic blood pressure (DBP), percutaneous coronary intervention (PCI). The non-invasive brachial measurements presented in Table 1 are those recorded at the time of the angiogram.

In total, 191 patients undergoing cardiac catheterization for clinical reasons were included in the study. According to the criteria of the 2013 AAMI protocol, 14 patients were excluded due to high variation of invasive BP readings (five patients with standard deviation (SD) in invasive systolic BP (SBP) >10 mmHg, three patients with SD in invasive mean arterial pressure (MAP) >6 mmHg and six patients with SD in invasive diastolic BP (DBP) >6 mmHg). Also, 22 patients were excluded due to severe arrhythmia during the measurements. In one study center, the first 10 measurements had to be excluded due to a systemic error in zeroing. This resulted in 145 patients being available for the analysis. All patients who entered the analysis were Caucasians. 107 patients were within a heart rate of 60–100/min (73.8%).

Table 2 shows the invasive blood pressure values and their distribution according to suggested blood pressure ranges.

*Study centers:* The measurements were performed in three German study centers: Greifswald, Bad Oeynhausen, and Bad Berka (for details please see the affiliations of 1, 3, and 4).

Most patients were included in Bad Berka (63 patients), followed by Bad Oeynhausen (50 patients) and Greifswald (32 patients). In all cases, the invasive recording was performed simultaneously to the non-invasive recording. Out of the 145 patients included in the analysis, 100 cardiac catheters were performed radially, and 45 were done femorally. The study was conducted in compliance with the Declaration of Helsinki. Ethics approval was obtained from the local ethics committees. All participants gave their informed consent for inclusion before they participated in the study.

*Study setting*: All measurements were performed in the cardiac catheterization laboratory, with constant temperature, without excessive ambient noise from monitoring devices. The invasive and non-invasive measurements were performed at the end of each cardiac catheter examination and were completely simultaneous. Thus, the patient was adapted to the environment and without disturbing influences. Data acquisition was done in a period of undisturbed rest, free from acute hemodynamic interventions, free from acute medication changes and without talking.

*Non-invasive central BP device measurement:* All non-invasive measurements were performed with the custo screen 400 device (custo med GmbH, Ottobrunn, Germany), with the integrated Antares algorithm to calculate central BP on a connected laptop. The familiarization with the equipment took only a few minutes in each study center because it is the process of a normal blood pressure measurement. The oscillometric device in its original version is built for 24 h-ambulatory blood pressure monitoring. The custo screen 400 is validated for peripheral blood pressure measurement according to ESH-IP 2010 [11]. Cuff size was chosen according to the directives given by the manufacturer after measurement of the circumference of the upper arm (small 20–24 cm, medium 24–32 cm, large 32–40 cm). The cuff was placed at the left arm, except for two patients (1%) in which the right upper arm was used. After placing the cuff at the upper arm, a first non-invasive measurement was performed before the cardiac catheterization in order to test the functioning of the device and to familiarize the patient with the measurement procedure. This measurement was not included in any calculations. The second measurement was done simultaneously with the invasive measurement on the opposite arm. In 99% of the cases invasive radial access was on the right site and the cuff for non-invasive recordings was placed on the left arm. It was started exactly after the recording of the catheter in the aorta ascendens was started. Time points of start and end of the non-invasive measurements were harmonized with the invasive measurement by setting time stamps and marks to definitely be sure of simultaneous recordings. When severe arrhythmia occurred during the non-invasive and simultaneous invasive measurement, a second recording was performed. If the second recording was disturbed by severe arrhythmia again, these recordings were not included in the analysis. After removing the unacceptable invasive measurements (please see below), all non-invasive measurements could be analyzed without having to discard a single measurement.

The Antares software version 2.0, developed by Redwave Medical GmbH, Jena, Germany, was applied in the oscillometric device. Acquisition of the oscillometric pulse waves took place during the deflation of the cuff. Cuff deflation speed was 4 mmHg/s with a linear deflation via a regulated valve. Redwave Medical is patent holder for pulse wave analysis (PWA) in pulse waves that are recorded during inflation and deflation of a cuff (patent no DE 10 2017 117 337 B4). Generally speaking, it means that the pulse waves generated during a normal oscillometric BP measurement procedure can be taken for PWA with no need for altering the standard BP pump operation. The recorded pulse waves were analyzed for non-invasive estimation of central BP using the Antares algorithm. In order to be independent of the potential error of peripheral BP measurement, a recalibration of the brachial BP waveforms was performed by the Antares algorithm internally. The recalibrated mean arterial pressure and diastolic pressure were used for calibration as an internal preprocessing step. The Antares algorithm receives a cuff pressure signal in deflation stage as input signal and separates the pulsatile signal component from the inherent cuff pressure. The single pulse waves are identified. Weighted, multiple transformation of each pulse wave is based on several analytical steps, which could be referred to as adaptive transfer function. Grid points are then identified to calculate hemodynamic parameters such as cBP. The residuum, defined as spread between actual and expected deflating cuff pressure, is calculated. Arrhythmia and other disturbing artifacts are identified based on the residuum and the shape of the pulse wave. The integration of Antares in the software of a blood pressure monitor aims to enable a brachial cuff-based BP monitor to be a type II-device with relatively accurate absolute central BP values independently from the peripheral BP measurement.

*Invasive central BP measurement:* The invasive central BP measurements and the non-invasive measurements were performed by following exactly the same protocol with one exception: the invasive recording time in Greifswald was 20 s, while in Bad Oeynhausen and Bad Berka it was 90 s. All invasive measurements were performed using fluid-filled catheters. In the majority of measurements (90%), a 5 French Judkins right or left 3.5 standard diagnostic coronary catheter of 100 cm length was used. A mix of multiple catheters was used in the remaining 10%. The test to determine frequency response involves a rapid flush like a rectangular pulse (at least 180 mmHg). After a sudden release, the flush bag pressure decreases rapidly and forces an overshooting of the baseline. The natural frequency was calculated from the time between 2nd and 3rd oscillation (one cycle). The damping coefficient was calculated from the ratio of the amplitudes of those two consecutive oscillations. The routinely checked natural frequency and damping coefficients of the systems were 21.9 Hz (15–29 Hz) and 0.21 (0.14–0.29), respectively, which surpasses the recommended guidelines [12]. The dataset used to develop the Antares algorithm is a different dataset from that used in the present validation. The full dataset of this study is for validation purposes only.

Flushing was performed before each invasive measurement using sodium chloride 0.9%. At the beginning of the invasive procedure, zeroing was performed precisely, having in mind that inaccuracy in zeroing will cause severe BP aberration. For calibration of each transducer the zero reference level for pressure measurement was set at midchest height, which was also used for balancing. Both calibration and balancing were checked before each measurement was performed. In an undefined number of cases repeated zeroing was performed according to the examiner’s experience. The correct catheter position was confirmed by X-ray at the end of the standard procedure, because the invasive measurement was performed at the end of each heart catheterization. Sample rates at which waveforms were invasively recorded were 500 Hz in Greifswald, 2000 Hz in Bad Oeynhausen, and 240 Hz in Bad Berka. Waveform data processing was performed with the use of a Philips Allura Xper FD20 system in Greifswald, Siemens Sensis Axiom system in Bad Oeynhausen and General Electric MacLab IT system in Bad Berka.

The invasive pressure waves were analyzed semi-automatically over the whole period of the recording; meaning 20 s in Greifswald, and 90 s in Bad Oeynhausen and Bad Berka, respectively. To determine the invasive systolic BP, the peak of every recorded pulse wave; for invasive diastolic BP the lowest signal point; and for invasive mean arterial pressure (MAP) the area under the curve was taken for the calculation. The pulse waves of the invasive recordings were visually checked and in that case cleared if they differed greatly from the mean (e.g., artifacts). As an example please see Figure 2a where, over 90 s, a total of two pulse waves had to be excluded. All other pulse waves were included in the analysis of central BP, what means that in the end several extrasystoles still were included. A measurement was considered severe arrhythmia if more than 30% of pulse waves had to be deleted, and the recording of that patient was withdrawn from the analysis. The values for each pulse wave were averaged and additionally the standard deviation was calculated. According to the AAMI protocol, recordings with a SD—within the invasive pressure—of more than 10 mmHg for systolic BP, 6 mmHg for diastolic BP, and 6 mmHg for MAP were withdrawn from the analysis. 

Nitroglycerin was injected at the beginning of the heart catheterization with a time difference of at least 10 minutes until the invasive and non-invasive recordings of the pulse waves were performed. No other medication was given closely prior to the recordings of the pulse waves.

Figure 1, Figure 2 and Figure 3 show original non-invasive and invasive recordings of a 65-year-old lady to illustrate how data were processed and analyzed.

All patient data and measurement results were stored in a database (Excel 2016, Microsoft Corp., Redmond, WA, USA). Statistical analysis was performed using IBM SPSS 22 software (IBM Corp., Armonk, NY, USA). Data are presented as means ±standard deviation. The Pearson correlation coefficient was used to assess the strength of linear correlation between invasive and estimated central BP. Furthermore, the regression equation y = ax + b was used to create a trend line in the scatter plots to evaluate the relationship between the pressures. The agreements between invasive and estimated central BP were compared using Bland–Altman analysis. Statistical significance was declared at the two-side *p* < 0.05 level.

## 3. Results

The coefficient of determination of estimated central systolic BP to invasively measured central systolic BP was *r*^2^ = 0.86. The mean difference was 0.71 mmHg and SD was 5.95 mmHg. For diastolic BP, the correlation was *r*^2^ = 0.71, mean difference 2.96 mmHg and SD 5.21 mmHg. For MAP, correlation was *r*^2^ = 0.84, mean difference 0.19 mmHg and SD 3.78 mmHg. The corresponding scatter plots and Bland–Altman plots revealed good limits of agreement. The trend lines illustrate no significant over-or underestimations for systolic BP, MAP or diastolic BP. Figure 4, Figure 5, Figure 6, Figure 7, Figure 8 and Figure 9 show the results in detail.

The mean standard deviation of the invasive central systolic blood pressure for all 145 patients was 4.08 mmHg (SD 1.62), for central mean arterial pressure 2.83 mmHg (SD 1.19), and for central diastolic blood pressure 2.31 mmHg (SD 1.06 mmHg). As stated above, patients with high invasive blood pressure SD were excluded from the analysis. Table 3 shows the distribution of measurement errors of the Antares algorithm within the ranges of <5 mmHg, <10 mmHg and <15 mmHg.

## 4. Discussion

The study confirmed that the Antares algorithm turns the oscillometric custo screen 400 blood pressure monitor into a type II-device for non-invasive estimation of true central BP fulfilling entirely the 2017 ARTERY validation protocol (Table 4) as well as the 2013 ANSI/AAMI/ISO 81060-2 and the 2018 AAMI/ESH/ISO validation protocol including the criteria for high-accuracy devices.

This validation study, which is comparing the invasively measured central BP with the non-invasive estimation using a regular oscillometric BP device, shows a level of agreement that faces the highest requirements as defined as the mean value of differences within five mmHg and standard deviation of less than 8 mmHg [7]. We did not show the results of agreement with the invasively measured cBP values when the non-invasive pulse waves were calibrated by the invasively measured MAP and DBP. The reason is that we strongly believe in clinical demands. The widespread use of invasive data for calibration of non-invasive oscillometric devices may lead to over-expectation of the performance of any oscillometric solution. However, clinically it would not make any sense to calibrate invasively a non-invasive oscillometric measurement for estimation of central BP. On the other hand, the good agreement with the invasive BP is achievable only if the peripheral BP is recalibrated to avoid the potential error of peripheral BP measurement as described earlier [8,12]. For recalibration, the area under the curve of every oscillometrically recorded pulse wave is used by Antares. The results of the recalibration are not displayed because they are for preprocessing purposes only. In other words: the Antares algorithm must have access to the oscillometric raw data in the deflation of the upper arm cuff to perform pulse wave analysis (PWA) with recalculation of MAP and diastolic BP as well as performing PWA for calculation of the corresponding central BP. Doing this, for systolic BP a mean difference of 0.71 mmHg is reached in comparison to the simultaneous invasive measurement.

According to the 2013 ANSI/AAMI/ISO 81060-2 protocol, the ranges of the invasively measured BP as reference blood pressure for comparison of the estimated central BP were for central systolic BP 10 mmHg and for diastolic BP 6 mmHg. In this way, a hemodynamic stability can be assumed. The residual pressure fluctuations may reflect a physiologic pressure range. Patients with arrhythmia in more than 30% of the recorded pulse waves were excluded. Again, this procedure is undertaken in order to achieve acceptable hemodynamic stability. To set the maximum for arrhythmic pulse waves to 30% of all invasively recorded pulse waves means that a relatively high number of patients with a certain degree of arrhythmia were still included in the calculation. Nevertheless, the level of agreement with a mean difference for estimated systolic BP of 0.71 mmHg and SD of 5.95 mmHg could be reached. Moreover, the data are pooled from three different study sites that could bear another potential error. Still, the measured BP is within the best acceptable range. “Best acceptable range” means for the 2017 ARTERY standard that the device tested passes the validation criteria with mean difference < 5 mmHg, SD < 8 mmHg without fail criteria, for the 2017 AAMI protocol that mean difference is below 5 mmHg with SD below 8 mmHg, for the 2018 AAMI/ESH/ISO protocol if an estimated probability of a tolerable error (≤ 10 mmHg) is at least 85%, and for the British Hypertension Society (BHS) if it passes grade A criteria. Consequently, for all named protocols, the results of the Antares algorithm to estimate cBP are within the best acceptable range. Thus, when implemented into the device, it may be concluded that the Antares algorithm is robust and could be used in the real world in order to non-invasively estimate the central BP.

### Limitations

All invasive measurements were performed using fluid-filled catheters. The biggest advantage of using fluid-filled catheters is that they can be easily integrated into everyday clinical practice. This allows achieving a sufficient number of patients included in the invasive measurement, is cost-effective and, furthermore, enhances the probability to show results that can be verified by other working groups easily. One disadvantage compared to micromanometer-tipped catheters is that they are susceptible to false pressures due to damping and altered frequency response. For the given fluid-filled catheters, the coefficient of damping and frequency response were within a generally accepted range [13]. Another advantage of high-frequency, micromanometer-tipped catheters with high-frequency acquisition systems is that they can be used to determine specific waveform features. Thus, the present validation focuses on the invasive validation of cBP and is not a validation of any waveform feature.

The invasive and non-invasive measurements were performed at the end of each cardiac catheter simultaneously. There was no lag of time between those measurements. However, 100% synchronization via R-peak was not done. To the best of our knowledge this should be acceptable because following the 2013 AAMI protocol, physiologic pressure fluctuations already were taken into account and should be covered by the length of the invasive measurements (20 s in Greifswald, 90 s in Bad Oeynhausen and Bad Berka, respectively). It cannot be completely ruled out that this procedure introduces a systematic error when comparing periods of different length. From a clinical point of view it may be advantageous to compare the estimated cBP with the invasive measurement procedure that comes as close as possible to the “true” central pressure. To average the cBP of an invasive recording of 90 s might be closer to the “true” cBP than a shorter period of 20 or 30 s. Therefore we followed the 2013 AAMI protocol and used the concept of hemodynamic stability of an entire invasive measurement, which timewise envelops the oscillometric measurement.

Though the development of the recalculated “new” brachial BP was done based on invasive comparisons (simultaneous invasive and non-invasive brachial BP measurements on both arms), the method of recalculation of the brachial BP is not invasively validated itself, but considered preprocessing only. Accordingly, the recalculated values are not displayed as discussed above. Thus, it has to be concluded that an exact estimation of aorta-to-brachial systolic BP amplification cannot be gauged using the Antares algorithm. This is partly because it is the nature of a type II-device instead of providing a type I-device. The decision to have a type II-device was reinforced by findings that type II-devices have proved better risk stratification for hypertension-related end organ damage and outcome then type I-devices [14,15,16,17,18].

The AAMI protocols as well the ARTERY protocol suggest that an indicative range for invasive central systolic BP may be below/equal 100 mmHg in 5% of the readings and above/equal 160 mmHg in 5% of the readings, and invasive central diastolic BP may be below/equal 60 mmHg in 5% of the readings and above/equal 85 mmHg in 5% of the readings. In the validation protocols this BP criterion is categorized as “may”, which means that this criterion provides further guidance and is not a “must” or “should” recommendation. Although we have measured invasively in total 191 patients and could use data of 145 patients for the analysis, we could include in the extreme BP ranges only three patients for central SBP below/equal 100 mmHg and 17 patients for central DBP of above/equal 85 mmHg. The corresponding Bland–Altman plots have a good level of accordance without any significant trend. However, it has to be stated that further invasive comparisons have to show if within these extreme BP ranges Antares works to a similarly robust degree as shown for the other ranges.

## 5. Conclusions

With the integrated Antares algorithm, the oscillometric blood pressure device studied here proofs to serve as a type II-device for estimation of true central BP. Antares fulfills entirely the 2017 ARTERY validation protocol as well as the 2013 ANSI/AAMI/ISO and the 2018 AAMI/ESH/ISO validation protocols including the criteria for high accuracy devices for estimation of central BP. Whether Antares can be integrated reliably with similar robust results in other oscillometric devices has yet to be proven.

The integration of a feature for measuring central blood pressure in commercially available blood pressure monitors could make it possible to measure these parameters in every practice in the future. The existing knowledge about the significance of central blood pressure could then finally be applied more widely in clinical practice.

## Figures and Tables

**Figure 1 jcm-08-01073-f001:**
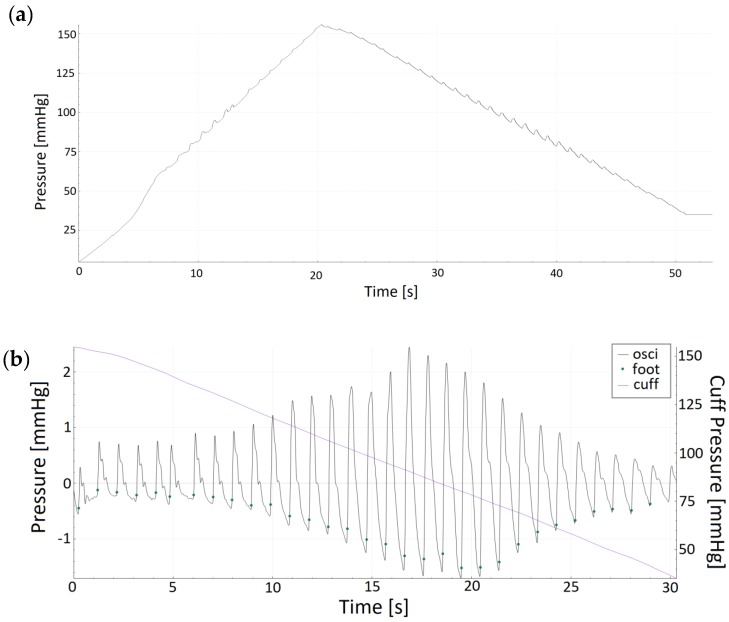
(**a**) Original oscillometric recording of a 65-year-old lady showing raw data during inflation and deflation of the oscillometric cuff. (**b**) Extracted oscillometric pulse waves of the same 65-year old lady during deflation of the cuff. The following signals are displayed: cuff pressure (cuff), extracted oscillometrical pulse waves (osci), and foot points for each identified pulse wave (foot) over time in seconds (s).

**Figure 2 jcm-08-01073-f002:**
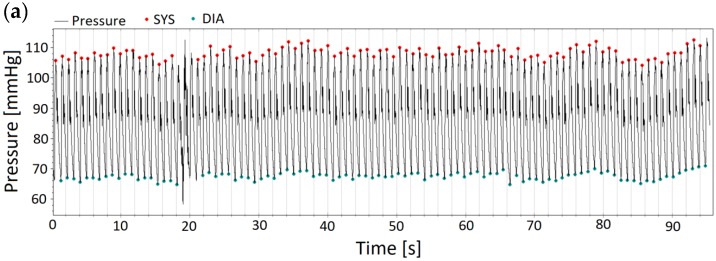
(**a**) Invasive raw data of the same 65-year-old lady. Over 90 s, a total of two pulse waves had to be excluded (at 18 s). All other pulse waves were included in the analysis of central BP. (**b**) Enlarged section of the invasive recording of pulse waves of the same 65-year-old lady. The points mark systolic and diastolic blood pressure.

**Figure 3 jcm-08-01073-f003:**
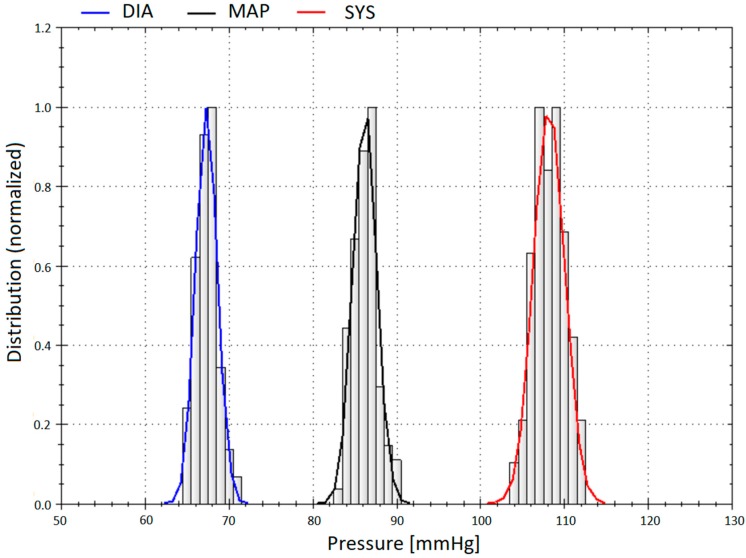
Distribution of the central blood pressures (BP) of the same 65-year-old lady, extracted from the invasive recordings with (from left to right) central diastolic BP, central mean arterial pressure, and central systolic BP.

**Figure 4 jcm-08-01073-f004:**
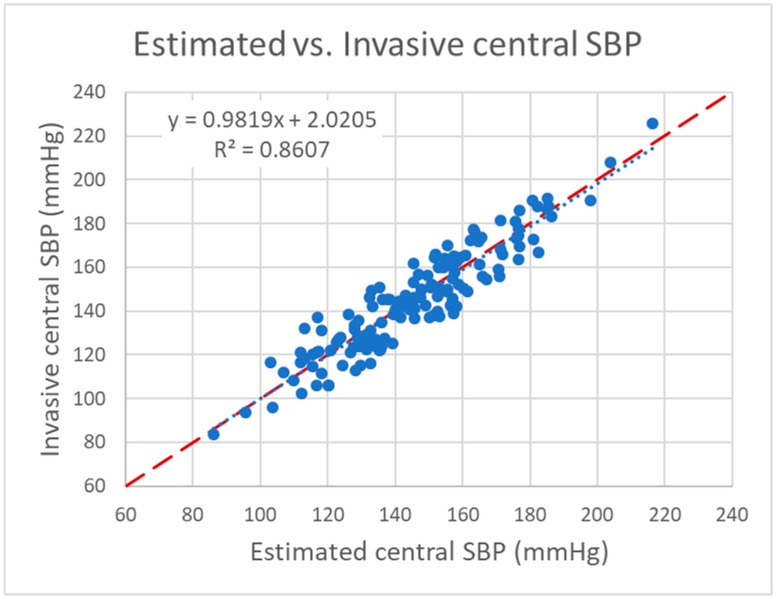
Scatter plot of estimated versus invasive central systolic blood pressure (SBP). Dashed line: line of identity; dotted line: trend line.

**Figure 5 jcm-08-01073-f005:**
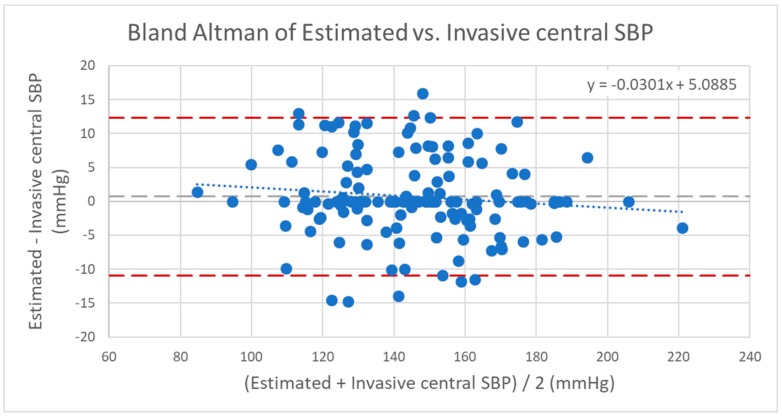
Bland–Altman of estimated versus invasive central systolic blood pressure (SBP). Dashed line (dark): confidence interval/upper and lower limits with mean ±1.96*standard deviation (SD); dashed line (light): mean value; dotted line: trend line of the scatter plot. Correlation of the coefficient of determination: *r*^2^ = 0.86; correlation: *r* = 0.93; mean: 0.71 mmHg; standard deviation: 5.95 mmHg; confidence interval 95%: −10.95/+12.37 mmHg (mean + 1.96*SD).

**Figure 6 jcm-08-01073-f006:**
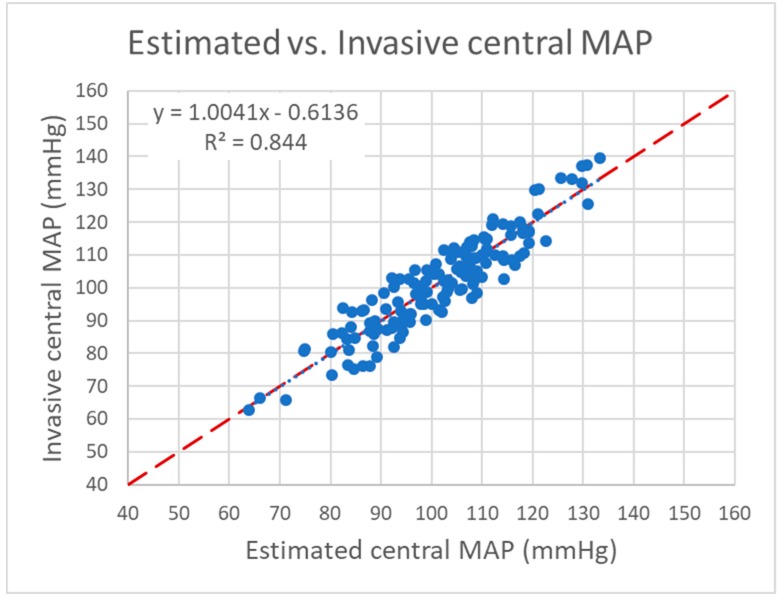
Scatter plot of estimated versus invasive central mean blood pressure (MAP). Dashed line: line of identity; dotted line: trend line.

**Figure 7 jcm-08-01073-f007:**
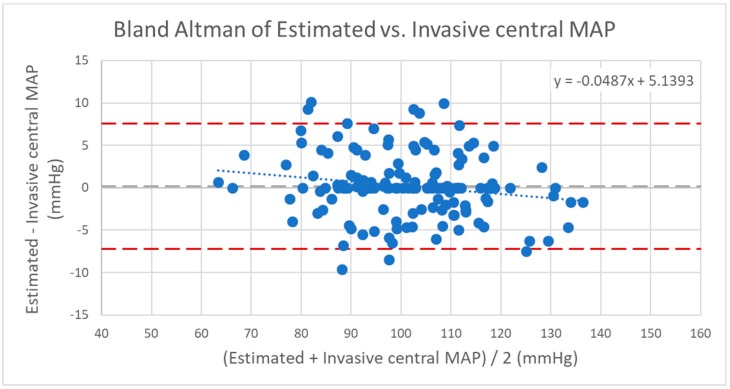
Bland–Altman of estimated versus invasive central mean blood pressure (MAP). Dashed line (dark): confidence interval/upper and lower limits with mean ±1.96*SD; dashed line (light): mean value; dotted line: trend line of the scatter plot. Correlation of the coefficient of determination: *r*^2^ = 0.84; correlation: *r* = 0.92; mean: 0.19 mmHg; standard deviation: 3.78 mmHg; confidence interval 95%: −7.21/+7.59 mmHg (mean + 1.96*SD).

**Figure 8 jcm-08-01073-f008:**
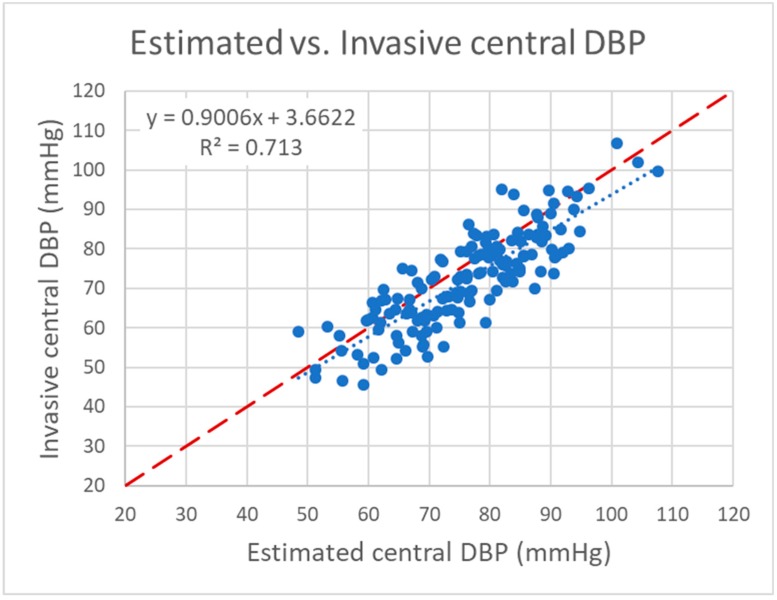
Scatter plot of estimated versus invasive central diastolic blood pressure (DBP). Dashed line: line of identity; dotted line: trend line.

**Figure 9 jcm-08-01073-f009:**
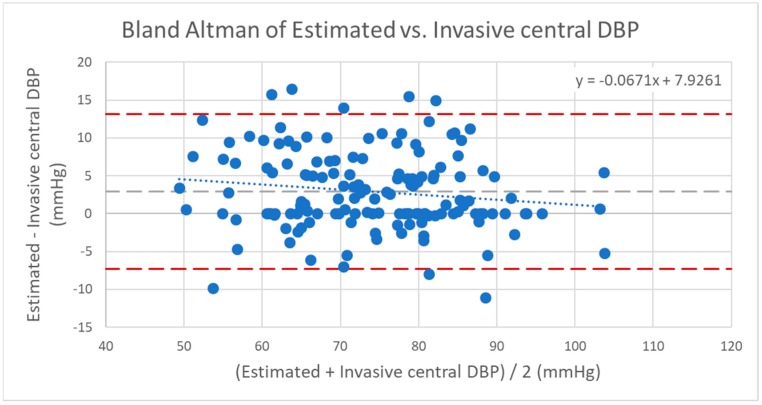
Bland–Altman of estimated versus invasive central diastolic blood pressure (DBP). Dashed line (dark): confidence interval/upper and lower limits with mean ±1.96*SD; dashed line (light): mean value; dotted line: trend line of the scatter plot. Correlation of the coefficient of determination: *r*^2^ = 0.71; correlation: *r* = 0.84; mean: 2.96 mmHg; standard deviation: 5.21 mmHg; confidence interval 95%: −7.25/+13.17 mmHg (mean + 1.96*SD).

**Table 1 jcm-08-01073-t001:** Patient characteristics.

	Mean	Min	Max	Standard Deviation
Number of included patients	145			
Male Sex	65.5%			
Age (years)	69.2	26	99	11.6
Height (cm)	171.7	115.0	195.0	10.0
Weight (kg)	83.8	34.0	129.0	17.3
Heart rate (1/min)	69.0	36.3	111.1	12.2
BMI (kg/m^2^)	28.4	16.6	64.3	4.6
Non-invasive brachial SBP (mmHg)	149.6	89.0	217.0	22.9
Non-invasive brachial MAP (mmHg)	107.4	64.0	151.0	16.6
Non-invasive brachial DBP (mmHg)	85.1	58.0	139.0	12.1
Invasive central SBP (mmHg)	145.1	83.4	225.7	24.0
Invasive central MAP (mmHg)	101.5	62.7	139.5	14.8
Invasive central DBP (mmHg)	72.1	45.5	106.7	12.2
No of patients with coronary artery disease	74 (51%)			
No of patients with diabetes	43 (30%)			
No of patients with hypertension	103 (71%)			
No of patients undergoing PCI	64 (44%)			
No of patients on antihypertensive medication	46 (32%)			

**Table 2 jcm-08-01073-t002:** Invasive blood pressures and their distribution according to ranges stated in ARTERY validation protocol and others. Central systolic blood pressure (cSBP), central diastolic blood pressure (cDBP).

	Number	% of Goal	ARTERY Goal
Number of included patients	145	171%	85
cSBP < = 100	3	2.1%	5%
cSBP > 100 < 140	58	40.0%	
cSBP > = 140	46	31.7%	20%
cSBP > = 160	38	26.2%	5%
cDBP < = 60	23	15.9%	5%
cDBP > 60 < 85	105	72.4%	
cDBP > = 85	15	10.3%	20%
cDBP > = 100	2	1.4%	5%

**Table 3 jcm-08-01073-t003:** Distribution of measurement errors of the Antares algorithm within the ranges of <5 mmHg, <10 mmHg and <15 mmHg. According all recommended standards, including Association for the Advancement of Medical Instrumentation (AAMI), ARTERY, European Society of Hypertension (ESH) and British Hypertension Society (BHS), Antares surpasses grade-A-criteria for central systolic blood pressure (SBP), central mean arterial pressure (MAP), and central diastolic blood pressure (DBP).

Error	<5 mmHg		<10 mmHg		<15 mmHg	
Estimated central SBP	89	61.4%	123	84.8%	144	99.3%
Estimated central MAP	118	81.4%	143	98.6%	145	100.0%
Estimated central DBP	89	61.4%	127	87.6%	141	97.2%

**Table 4 jcm-08-01073-t004:** Summary of central blood pressure device validation protocol components and requirements of the 2017 ARTERYs validation consensus statement applied to the present validation study.

Protocol Section	Protocol Item	Protocol Requirement	Protocol Undertaken with Comment
Study setting	Isolated room without disturbing influences	Should	Yes
Non-invasive central blood pressure device measurement standards	List manufacturer, model, software version, operating principles, signal processing step/s, calibration processes	Must	Yes
	Time for blood pressure measures; time points of brachial and central blood pressure; cuff deflation speed	Should	Yes
	Define and use appropriate cuff size	Must	Yes
	Dimensions of inflatable bladder for all cuff sizes available; process to determine cuff size	Should	Yes
	Process familiarization with equipment	Should	Yes
	Separate validation studies for additional or optional features or functions	Must	Yes, here focus on central blood pressures
	Process of quality control; process used to delineate acceptable quality; number of unacceptable readings; reason/s for exclusion	Must	Yes
Invasive (intra-arterial) central blood pressure reference standard	Micromanometer-tipped catheter used if minor inflection points to be identified	Should	Not applicable because no waveform features are topic of this validation
	Full description of catheter; frequency response and handling procedures	Must	Yes
	Performance comparison of fluid-filled catheter with micromanometer-tipped catheter	May	No
Data acquisition at rest	Period of undisturbed rest; medications used	Should	Yes
	No talking. Free from acute hemodynamic interventions	Must	Yes
	Test device compared with reference over time-period matching the test device deflation cycle; recorded under stable conditions	Must	Yes
	Complete description of protocol; time interval between test device and reference measures	Must	Yes
Data acquisition at blood pressure intervention	Hemodynamic change from resting state	May	No
	Description of the intervention procedure	Must	Not applicable because no intervention done

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
