# Peer review of "Invasive Validation of Antares, a New Algorithm to Calculate Central Blood Pressure from Oscillometric Upper Arm Pulse Waves"

_jcm, 2019, doi:10.3390/jcm8071073_

Round 1

Reviewer 1 Report

The study by Dörr et al reports the invasive validation of an algorithm to estimate central blood pressure from an upper-arm oscillometric cuff BP device. The key findings were that the algorithm allows an accurate and non-invasive estimation of central aortic systolic and diastolic blood pressure, when compared against invasive aortic values. The authors should be commended for conducting a rigorous validation against internationally recognised protocols of this novel algorithm. Indeed, this is highly novel and impressive data. However, several issues need to be addressed by the authors.

Specific comments

1.       Most central blood pressure devices use a ‘generalised transfer function’ to transmute a peripheral blood pressure waveform to an estimate of the central blood pressure waveform. Additional detail on the Antares pulse wave analysis process is warranted in the Methods.

2.       In figure 1b, the oscillometric pulse waves vary greatly in amplitude, but these same waves are calibrated and used to generate the central blood pressure. Can more detail be provided on the processing of the waves (e.g. ensemble averaging or whether only specific sections of the oscillometric pulse are used for central BP).

3.       Figure 1b should be explained in more detail. There appears to be several different waveforms within the figure, including beat-to-beat estimation of the central aortic pressure? If this is the case, it is an important detail that should be explained in more detail.

4.       Following point 4, does the device generate a central aortic waveform and associated parameters (e.g. augmentation index)?

5.       Lines 172-3: “The invasive pressure waves were analyzed semi-automatically over the whole period of the recording; meaning 20s in Greifswald, and 90s in Bad Oeynhausen and Bad Berka, respectively”. Figure 1b is a recording over 30 seconds, does this mean that the oscillometric and invasive BP was not analysed over precisely the same time period? Wouldn’t it be more appropriate to analyse the precisely simultaneous period because time stamps/marks are available?

6.       Line 176: “Obviously false single beats (e.g. extrasystole, artifacts) were deleted.” Please be more specific with respect to the process used to define which beats were deleted and the prevalence of deletions. Please also describe if this was performed for both the invasive and oscillometric recordings.

7.       The 2017 ARTERY statement checklist should be included as an additional table (see Table 3 of Sharman et al, Eur Heart J. 2017;38(37):2805-12).

8.       Lines 306-308: “the method of recalculation of the brachial BP is not invasively validated itself, but considered preprocessing only.” This data should be reported in the Results section because it is important to the function of the device.

Minor

1.       Lines 262-264: ‘r2=0.94’. This is a result and should be placed in the results section or into an appendix if the authors wish to retain focus on the non-invasive calibration only. It should not be placed in the Discussion.

2.       Table 1: Please provide a measure of variance along with the mean (e.g. mean ± standard deviation). Please clarify if the non-invasive brachial measurements presented in table 1 are those recorded at the time of the angiogram.

3.       Table 2: The ARTERY document suggests device accuracy should be tested across a range of heart rates. It would be additive to include the number of subjects with heart rate 60-100 beats/minute.

Author Response

Open Review

(x) I would not like to sign my review report

( ) I would like to sign my review report

English language and style

( ) Extensive editing of English language and style required

(x) Moderate English changes required

( ) English language and style are fine/minor spell check required

( ) I don't feel qualified to judge about the English language and style

                Yes         Can be improved            Must be improved         Not applicable

Does the introduction provide sufficient background and include all relevant references?

                ( )           (x)          ( )           ( )

Response: We would be very grateful for any hint on how we can improve the introduction!

Is the research design appropriate?

                (x)          ( )           ( )           ( )

Are the methods adequately described?

                ( )           (x)          ( )           ( )

Response: We went through each of your comment/criticism, resulting in some changes in the methods. If you do have more ideas how to improve the methods further, again we would be very grateful. 

Are the results clearly presented?

                ( )           (x)          ( )           ( )

Response: We went through each of your comment/criticism, resulting in some changes in the results. If you do have more ideas how to improve the results further, again we would be very grateful!

Are the conclusions supported by the results?

                (x)          ( )           ( )           ( )

Comments and Suggestions for Authors

The study by Dörr et al reports the invasive validation of an algorithm to estimate central blood pressure from an upper-arm oscillometric cuff BP device. The key findings were that the algorithm allows an accurate and non-invasive estimation of central aortic systolic and diastolic blood pressure, when compared against invasive aortic values. The authors should be commended for conducting a rigorous validation against internationally recognised protocols of this novel algorithm. Indeed, this is highly novel and impressive data. However, several issues need to be addressed by the authors.

Specific comments

1.       Most central blood pressure devices use a ‘generalised transfer function’ to transmute a peripheral blood pressure waveform to an estimate of the central blood pressure waveform. Additional detail on the Antares pulse wave analysis process is warranted in the Methods.

Response 1: Thanks for drawing attention on this important point. The present manuscript is an invasive validation paper. Thus, none of the authors knows too many details of the algorithm. To address the point raised we contacted Chris Stockmann, one of the inventors of Antares, in order to provide more details about the algorithm. As a result of intensive discussion we added to the methods section:

“The Antares algorithm receives a cuff pressure signal in deflation stage as input signal and separates the pulsatile signal component from the inherent cuff pressure. The single pulse waves are identified. Weighted, multiple transformation of each pulse wave is based on several analytical steps, which could be referred to as adaptive transfer function. Grid points are then identified to calculate hemodynamic parameters such as cBP. The residuum, defined as spread between actual and expected deflating cuff pressure is calculated. Arrhythmia and other disturbing artifacts are identified based on the residuum and the shape of the pulse wave.”

Please allow us to add that to the best of our knowledge this is one of the most detailed descriptions of an algorithm within a validation paper. Thus, we hope that the Reviewer can follow the description and may accept the limit of information given according to the details of the algorithm. For more details the Reviewer may contact Redwave Medical directly.

Because Chris Stockmann is chief technology officer of Redwave Medical, we added to the acknowledgement a note what honestly we should have done before “We are grateful to Chris Stockmann (Redwave Medical GmbH) for providing information of the Antares algorithm and figures 1a and 1b”. Additionally we changed the conflict of interests section: ”Redwave Medical GmbH had no role in the design, execution, or interpretation of the study. Redwave Medical GmbH provided figures 1a and 1b, as well as detailed information about the algorithm requested by the reviewer.”

2.       In figure 1b, the oscillometric pulse waves vary greatly in amplitude, but these same waves are calibrated and used to generate the central blood pressure. Can more detail be provided on the processing of the waves (e.g. ensemble averaging or whether only specific sections of the oscillometric pulse are used for central BP).

Response 2: The answer is for the most part already given in response 1. Additionally, we added to the method sections: “The entire cuff pressure signal is used for calculation, in order to get robustness against natural physiologic shifts of BP within the measurement time as good as possible.” Figure 1b has included the information that every single pulse wave recorded during deflation of the cuff is analyzed, what again is stated in the methods section.

We have exchanged the figures 1a and 1b where German words were exchanged by English words. We do apologize for that sloppiness. The same is true for the figures 2a, 2b and 3. Moreover we changed the legends of Figures 4, 6 and 8, legend: “Dashed line: 45° reference” was replaced by “Dashed line: line of identity” and in Figures 5/7/9 x-axis numbers are now displayed below (not inside) plot.

3.       Figure 1b should be explained in more detail. There appears to be several different waveforms within the figure, including beat-to-beat estimation of the central aortic pressure? If this is the case, it is an important detail that should be explained in more detail.

Response 3: We entirely agree with your comment and are thankful to give us the chance to correct this mistake. We have deleted Figure 1b and replaced it with a much clearer figure without unnecessary information like ECG. Accordingly, we have changed the legend of figure 1b and changed to English:

Figure 1b. Extracted oscillometric pulse waves of the same 65-year old lady during deflation of the cuff. The following signals are displayed: cuff pressure (cuff), extracted oscillometrical pulse waves (osci), and foot points for each identified pulse wave (foot) over time in seconds (s).

4.       Following point 4, does the device generate a central aortic waveform and associated parameters (e.g. augmentation index)?

Response 4: Yes, the algorithm generates central aortic pulse waves, which are used to determine other hemodynamic parameters, e.g. augmentation index and augmentation pressure. But this is not part of the present validation. One reason for this is that fluid filled catheters have been used which may not be sufficient to determine waveform features, in which minor inflection points need to be identified. Thus, for these waveform features a separate validation would be needed.

We added to the last part of the discussion section where the use of fluid-filled catheters is discussed: “Another advantage of high frequency, micromanometer-tipped catheters with high-frequency acquisition systems is that they can be used to determine specific waveform features. Thus, the present validation focuses on the invasive validation of cBP and is not a validation of any waveform feature.”

5.       Lines 172-3: “The invasive pressure waves were analyzed semi-automatically over the whole period of the recording; meaning 20s in Greifswald, and 90s in Bad Oeynhausen and Bad Berka, respectively”. Figure 1b is a recording over 30 seconds, does this mean that the oscillometric and invasive BP was not analysed over precisely the same time period? Wouldn’t it be more appropriate to analyse the precisely simultaneous period because time stamps/marks are available?

Actually during the conceptualization process of the study we were discussing this point intensively. We came to the conclusion that we might follow the 2013 AAMI protocol. AAMI has included criteria that should guarantee hemodynamic stability. We do have invasive measurements of 90 s length in two of the three study sites. This gives us the chance to define the hemodynamic stability of the patient quite well and come close to the “true” central BP of this particular patient. The longer the period of invasive recording, the closer one gets to the “true” central BP. From a clinical point of view we are interested in “true” central pressure. In a clinical setting beat-to-beat-synchronization would be non-sense. And we really believe in clinical demands. Thus, we decided to compare the non-invasive acquisition of the oscillometric pulse waves with the longest possible period of invasive recordings to come close to the “true” central pressure.  In our opinion another advantage in following the AAMI protocol is that if other groups perform invasive measurements and compare their results with Antares, their results should be closer to our results even if they do not have exact simultaneous measurements. We may have better correlations in the present study if we analyze the precisely simultaneous period, but we may be farer away from the “true” cBP of the patient. Additionally we have understood the 2017 ARTERY statement in that way that this would be acceptable. The statement claims that the “non-invasive central BP values must be compared with the intra-arterial central BP (reference) values averaged over a time-period matching the deflation cycle of the non-invasive device and recorded under stable conditions, ideally simultaneously, or as contemporaneous as possible.” We definitely fulfill this criterion with having the non-invasive measurement covered by the invasive recordings in parallel. The ARTERY statement even would allow a comparison of invasive and non-invasive measurements with some additional requirements. Thus, during conceptualization process, we decided to go that way, we believe that this is reliable and even do hope more that you may can accept our thoughts.

In the limitations section we have described that the oscillometric recordings are 100% within the invasive recordings (examination order: invasive recording started, oscillometric recording started, oscillometric recording finished, invasive recording finished) and that we did not do a beat-to-beat-synchronization. Please allow us to give further explanation if required to the already given information in the limitation section. We added the limitations section:

“The invasive and non-invasive measurements were performed at the end of each cardiac catheter simultaneously. There was no lag of time between those measurements. However, a 100% synchronization via R-peak was not done. To the best of our knowledge this should be acceptable because following the 2013 AAMI protocol, physiologic pressure fluctuations already were taken into account and should be covered by the length of the invasive measurements (20s in Greifswald, 90s in Bad Oeynhausen and Bad Berka, respectively). From a clinical point of view it may be advantageous to compare the estimated cBP with the invasive measurement procedure that comes as close as possible to the “true” central pressure. To average the cBP of an invasive recording of 90s might be closer to the “true” cBP than a shorter period of 20 or 30 seconds. The study should evaluate robustness and reliability for use of Antares in real life scenarios, where R-peak synchronization is not done. Therefore we followed the 2013 AAMI protocol and used the concept of hemodynamic stability of an entire invasive measurement, which timewise envelops the oscillometric measurement.”

6.       Line 176: “Obviously false single beats (e.g. extrasystole, artifacts) were deleted.” Please be more specific with respect to the process used to define which beats were deleted and the prevalence of deletions. Please also describe if this was performed for both the invasive and oscillometric recordings.

Response 6: We exchanged Fig 2a and added information of deleted pulse waves. Moreover we added in the methods section: “The pulse waves of the invasive recordings were visually checked and in that case cleared if they differed greatly from the mean (e. g. artifacts). As an example please see Fig 2a, where over 90 s altogether two pulse waves had to be excluded. All other pulse waves were included in the analysis of central BP, what means that in the end several extrasystoles still were included.”

This was performed solely for the invasive recordings. If Antares has excluded single pulse waves from analysis, this is unknown to the authors because we just took the result of calculation of the estimated cBPs.

7.       The 2017 ARTERY statement checklist should be included as an additional table (see Table 3 of Sharman et al, Eur Heart J. 2017;38(37):2805-12).

Response 7: We added the ARTERYs checklist as table 4

Within the process of checking again the validation protocol components and requirements we got aware that we had to add another two sentences in the methods section: “The familiarization with the equipment took only a few minutes in each study center because it the process of a normal blood pressure measurement.” and “After removing the unacceptable invasive measurements (please see below), all non-invasive measurements could be analyzed without having to discard a single measurement.”

Thank you again for the note to insert this table with the corresponding cross check.  

8.       Lines 306-308: “the method of recalculation of the brachial BP is not invasively validated itself, but considered preprocessing only.” This data should be reported in the Results section because it is important to the function of the device.

Actually because this is a validation study of the estimated central BP and not of any preprocessing steps the authors are not familiar with, we rather would not show any results of preprocessing steps. With the help of Chris Stockmann we could get some details of the algorithm that are written down in the discussion section (please see Response 1). This is an invasive validation of the central BP and not of other variables that might be calculated by Antares (also see response 4). Thus, we deleted ‘r2=0.94’.

Minor

1.              Lines 262-264: ‘r2=0.94’. This is a result and should be placed in the results section or into an appendix if the authors wish to retain focus on the non-invasive calibration only. It should not be placed in the Discussion.

Response 1: We fully agree and deleted ‘r2=0.94’. Yes, indeed, we do not want to calibrate with the invasive pressure, because this would be clinically senseless – as stated in the discussion section.

2.       Table 1: Please provide a measure of variance along with the mean (e.g. mean ± standard deviation). Please clarify if the non-invasive brachial measurements presented in table 1 are those recorded at the time of the angiogram.

Response 2: We added to table 1 the standard deviation. We added to the legend of table 1: “The non-invasive brachial measurements presented in table 1 are those recorded at the time of the angiogram.”

3.       Table 2: The ARTERY document suggests device accuracy should be tested across a range of heart rates. It would be additive to include the number of subjects with heart rate 60-100 beats/minute.

Response 3:  We added to the methods section: “107 patients were within a heart rate of 60-100/min (73.8%).”

Reviewer 2 Report

Dörr et al. present a validation study of a novel algorithm to calculate central blood pressure. Although the study is interesting, I have some comments.

MAJOR
Antares is designed to develop type-II devices, or, in other words, to facilitate measurement of absolute central blood pressure. This implies that the algorithm has to do two things: 1) properly describe the actual pressure amplification (like a type-I device should), but also 2) correct for the difference between an invasive and non-invasive (cuff) blood pressure measurement. In the case of Antares, both are integrated into one algorithm. Step 2, however, is potentially very device-dependent. Hence, I feel that this study is not “validating Antares”, but instead “validating Antares in the Custo Med device”; or, if you will, “validating the Custo Med device as a type II blood pressure measurement device”. This is an important distinction and limitation. An *algorithm* cannot be “type II” – only a device can.
Because Antares includes *both* steps as described above, most probably, tailored Antares algorithms will be needed for different devices. This distinction should be clear throughout the manuscript. E.g., I suggest re-phrasing the title.
Please comment.

Which dataset was used to develop the Antares algorithm? The same dataset as presented here? For proper validation, these should be completely independent.

The authors collected a very rich data set. However, as laid out in line 301: “a 100% synchronization via R-peak was not done.” Why not? This would considerably strengthen this validation study. Please perform this analysis.

MINOR
General (detailed examples below, e.g.: “the highest validation criteria”, “good limits of agreement”, “as precisely as possible”, “obviously”): please be explicit, quantitative, and objective. This is a scientific manuscript – not a sales flyer!

Abstract, line 25: “Invasive cBP recordings were prospectively compared to simultaneous non-invasive cBP estimations using the Antares algorithm, integrated into an oscillometric BP monitor.” Please explain the use of the term “prospectively” here.

Abstract, line 28: “fulfilling the highest validation criteria”. Please be explicit about which criteria.

Abstract, line 29: “Bland-Altman plot reveals good limits of agreement.” Please rephrase and quantify.

Abstract, lines 29-31: “Antares is the first algorithm for estimation of cBP that entirely fulfills the 2017 ARTERY and AAMI/ESH/ISO validation protocols including criteria for high accuracy devices.” See major comment #1 – the guidelines mentioned are for *device* validation. The present study shows that Antares works well *in a particular device*, which cannot be generalized to other devices.

Table 1: Min and Max only represent outliers. Please report median, 25th and 75th percentile values, or mean and standard deviation.

Table 2, first line: “% of goal”: this is only correct for the number of patients, for the other lines it represents % of patients

Lines 106-110 refer to protocol differences, but the protocol has not been explained here yet. Please move down.

Line 111: radial catheters: were these in the same arm that was used for brachial BP estimation? If so, does this influence the brachial BP measurement?

Line 126/127: I would suggest excluding the two patients in whom the right arm was used.

Lines 162-164: “zeroing was performed as precisely as possible”. What does that mean? How was the transducer height chosen?

Line 176: delete “obviously”

All figures: Please use English. Numbers, lines and labels should be much larger for readability.

Figure 1: please align x-axes of the two panels, and use the same x-axis for panel 2a

Figure 1b: multiple lines are plotted. What do they represent? Please use different colors/line types (e.g. dashing). Please also include an extra panel that shows one or two beats zoomed in, so we can appreciate the differences.

Figure 2a. this is nice, but what’s relevant is the beats that were recorded simultaneous with the brachial beats.

Please provide a figure where brachial and central blood pressure waves are plotted together in the same panel, so that waveform differences can be appreciated.

Figures 4-9: please combine into one figure.

Figure 4/6/8 legend: “Dashed line: 45° reference”. Please replace with “Dashed line: line of identity”.

Figures 5/7/9: please display x-axis numbers below (not inside) plot.

Line 255 “the oscillometric blood pressure device”. Please replace with “the Custo Med Custo Screen 400 device”.

Line 259 Please delete “prospective”.

Line 286: “Moreover, the data are pooled from three different study sites what could bear another potential error.” Please compute Bland-Altman statistics for these studies separately and compare the results (can be in a supplemental material).

Line 287 “within the best acceptable range”. Please reformulate.

Lines 290-297: I suggest moving these to the limitations section.

Line 300: “lack” -> “lag”

Lines 300-301: “There was no lack of time between those measurements. However, a 100% synchronization via R-peak was not done.” These two sentences contradict each other. Why wasn’t an R-wave synchronization performed?

Line 326 “blood pressure device”: please add “blood pressure device studied here”

Lines 339-342: No funding information is provided. Please provide funding information.

Author Response

Response to Reviewer 2 Comments

Open Review

(x) I would not like to sign my review report

( ) I would like to sign my review report

English language and style

( ) Extensive editing of English language and style required

(x) Moderate English changes required

( ) English language and style are fine/minor spell check required

( ) I don't feel qualified to judge about the English language and style

                Yes         Can be improved              Must be improved           Not applicable

Does the introduction provide sufficient background and include all relevant references?

                ( )            (x)           ( )            ( )

Response: We would be very grateful for any hint on how we can improve the introduction!

Is the research design appropriate?

                (x)           ( )            ( )            ( )

Are the methods adequately described?

                ( )            (x)           ( )            ( )

Response: We went through each of your comment/criticism, resulting in some changes in the methods. If you do have more ideas how to improve the methods further, again we would be very grateful. 

Are the results clearly presented?

                ( )            (x)           ( )            ( )

Response: We went through each of your comment/criticism, resulting in some changes in the results. If you do have more ideas how to improve the results further, again we would be very grateful!

Are the conclusions supported by the results?

                (x)           ( )            ( )            ( )

Comments and Suggestions for Authors

Dörr et al. present a validation study of a novel algorithm to calculate central blood pressure. Although the study is interesting, I have some comments.

MAJOR

Point 1: Antares is designed to develop type-II devices, or, in other words, to facilitate measurement of absolute central blood pressure. This implies that the algorithm has to do two things: 1) properly describe the actual pressure amplification (like a type-I device should), but also 2) correct for the difference between an invasive and non-invasive (cuff) blood pressure measurement. In the case of Antares, both are integrated into one algorithm. Step 2, however, is potentially very device-dependent. Hence, I feel that this study is not “validating Antares”, but instead “validating Antares in the Custo Med device”; or, if you will, “validating the Custo Med device as a type II blood pressure measurement device”. This is an important distinction and limitation. An *algorithm* cannot be “type II” – only a device can.

Because Antares includes *both* steps as described above, most probably, tailored Antares algorithms will be needed for different devices. This distinction should be clear throughout the manuscript. E.g., I suggest re-phrasing the title.

Please comment.

Response 1: Thank you for giving us the opportunity to further explain the principles standing behind the Antares algorithm. We appreciate very much to follow your demand to comment on the points raised by you above. We do it the best we can. But to some extend we have to refer to Redwave Medical because they developed the algorithm and may provide you with more information in detail. Sure, they provided us, the authors and the ones who performed the validation study with helpful information about the Antares algorithm before we started the present study. We are very much willing to offer this information. Actually you state that “This implies that the algorithm has to do two things: 1) properly describe the actual pressure amplification (like a type-I device should), but also 2) correct for the difference between an invasive and non-invasive (cuff) blood pressure measurement. In the case of Antares, both are integrated into one algorithm.” Additionally you state “Because Antares includes *both* steps as described above”. We hesitate to answer because you are the esteemed reviewer, but this is not true. Antares turns the custo screen 400 BP monitor very clearly into a type II device and not into a type I device. This is stated many times in the manuscript and described in the methods section in more detail. The process of recalibration is a step of pure preprocessing that is not displayed and not included in any report. This is an invasive validation of the central BP and not of other variables that might be calculated by Antares. If Antares is doing that, this would demand a separate validation study. The 2017 ARTERY statement defines type I and type II as follows: ”Type I—device purports to give an estimate of central BP relative to measured brachial BP (i.e. relatively accurate pressure difference between central and peripheral sites). Type II—device purports to estimate the intra-arterial central BP (i.e. relatively accurate absolute central BP value despite inaccuracy at the peripheral site).” This is why we have chosen to validate against the invasive intra-arterial pressure and not to the invasive pulse pressure amplification as it would have been necessary for an invasive validation of a type I device.  We are sorry but we do not see at all how Antares will be an algorithm for turning a BP monitor into a type I device. Thus, the deduction you stated might be subject of rethinking.

To the best of our knowledge, Antares was developed to be integrated in several regular blood pressure monitors. The custo screen 400 monitor is a regular 24h-ABPM monitor. When the Antares algorithm is integrated, then it can provide central blood pressures that are passing criteria from 2017 ARTERY statement, 2013 AAMI protocol, 2018 AAMI/ESH/ISO criteria and BHS grade A criteria (see responses 3, 4, 6 and 27), each with the highest defined criteria. Antares is a novel type of algorithm which has been designed device independent. Therefore, on one hand it requires access to the unfiltered raw data of any blood pressure cuff. On the other hand, it does not follow the operational steps of current devices of first handling pressure amplification and afterwards correcting the difference between invasive and oscillometric measurements (or vice versa). Consequently, since Antares is not performing those two steps, it does not need to be tailored for different devices. Nevertheless, it is clearly mentioned, that if Antares can be integrated reliably with similar robust results in other oscillometric devices has to be proven.  Please see limitations section: “If Antares can be integrated reliably with similar robust results in other oscillometric devices has to be proven.”

An excellent example of an invasive validation of an algorithm to estimate central BP is done by Dr. Thomas Weber (Weber T, Wassertheurer S, Rammer M, Maurer E, Hametner B, Mayer CC, Kropf J, Eber B. Validation of a brachial cuff-based method for estimating central systolic blood pressure. Hypertension 2011;58:825–832.). Together with the inventor of the ARCSolver he validated this algorithm that basically is doing the same like Antares. The main message is the same within the present paper, that ARCSolver is invasively validated, not just the oscillometric device. Even if ARCSolver runs in SphygmoCor and analyzes its pressure waves, no one would argue that this is a validation of the SphygmoCor device and not the ARCSolver algorithm. So according to the literature the present validation and interpretation seems to be in line with former publications of algorithm validations.

However, to address your criticism even more, we added in the abstract “The Antares algorithm turns the custo screen 400 BP monitor into a type II device.”

Which dataset was used to develop the Antares algorithm? The same dataset as presented here? For proper validation, these should be completely independent.

Response 2:  We fully agree. The authors are not aware of the dataset of which Antares was developed with. Our dataset is used solely for validation purpose. This is the reason, why we called it a “prospective” validation study, though we changed this term according to your comment/response 5. We added to the Methods section: “The dataset used to develop the Antares algorithm is a different dataset from that used in the present validation. The full dataset of this study is for validation purposes only.”

The authors collected a very rich data set. However, as laid out in line 301: “a 100% synchronization via R-peak was not done.” Why not? This would considerably strengthen this validation study. Please perform this analysis.

Response 3: Actually during the conceptualization process of the study we were discussing this point intensively. We came to the conclusion that we might follow the 2013 AAMI protocol, what is also accepted following the 2017 ARTERY statement (see below). AAMI has included criteria that should guarantee hemodynamic stability. We do have invasive measurements of 90 s length in two of the three study sites. This gives us the chance to define the hemodynamic stability of the patient quite well and come close to the “true” central BP of this particular patient. The longer the period of invasive recording, the closer one gets to the “true” central BP. From a clinical point of view we are interested in “true” central pressure. In a clinical setting beat-to-beat-synchronization would be non-sense. And we really believe in clinical demands. Thus, we decided to compare the non-invasive acquisition of the oscillometric pulse waves with the longest possible period of invasive recordings to come close to the “true” central pressure.  In our opinion another advantage in following the AAMI protocol is that if other groups perform invasive measurements and compare their results with Antares, their results should be closer to our results even if they do not have exact simultaneous measurements. We may have better correlations in the present study if we analyze the precisely simultaneous period, but we may be farer away from the “true” cBP of the patient. Additionally we have understood the 2017 ARTERY statement in that way that this would be acceptable. The statement claims that the “non-invasive central BP values must be compared with the intra-arterial central BP (reference) values averaged over a time-period matching the deflation cycle of the non-invasive device and recorded under stable conditions, ideally simultaneously, or as contemporaneous as possible.” We definitely fulfill this criterion with having the non-invasive measurement covered by the invasive recordings in parallel. The ARTERY statement even would allow a comparison of invasive and non-invasive measurements with some additional requirements. Thus, during conceptualization process, we decided to go that way, we believe that this is reliable and even do hope more that you may can accept our thoughts.

In the limitations section we have described that the oscillometric recordings are 100% within the invasive recordings (examination order: invasive recording started, oscillometric recording started, oscillometric recording finished, invasive recording finished) and that we did not do a beat-to-beat-synchronization. Please allow us to give further explanation if required to the already given information in the limitation section. We added the limitations section:

“The invasive and non-invasive measurements were performed at the end of each cardiac catheter simultaneously. There was no lag of time between those measurements. However, 100% synchronization via R-peak was not done. To the best of our knowledge this should be acceptable because following the 2013 AAMI protocol, physiologic pressure fluctuations already were taken into account and should be covered by the length of the invasive measurements (20s in Greifswald, 90s in Bad Oeynhausen and Bad Berka, respectively). From a clinical point of view it may be advantageous to compare the estimated cBP with the invasive measurement procedure that comes as close as possible to the “true” central pressure. To average the cBP of an invasive recording of 90s might be closer to the “true” cBP than a shorter period of 20 or 30 seconds. The study should evaluate robustness and reliability for use of Antares in real life scenarios, where R-peak synchronization is not done. Therefore we followed the 2013 AAMI protocol and used the concept of hemodynamic stability of an entire invasive measurement, which timewise envelops the oscillometric measurement.”

MINOR

General (detailed examples below, e.g.: “the highest validation criteria”, “good limits of agreement”, “as precisely as possible”, “obviously”): please be explicit, quantitative, and objective. This is a scientific manuscript – not a sales flyer!

Response 4: Thank you for providing us with the information that you might have understood our paper as being a sales flyer. We take this point very serious. On the one hand, it shows that the validation of the Antares algorithm leaves a strong impression fulfilling the highest validation criteria (also see responses 1, 3, 6, 7, 15 and 27). On the other hand, wherever our manuscript is not clearly scientific, it must be changed. We are grateful for detailed information where the information given by us is not clearly scientific; please see also responses 1, 3, 6, 7, 15 and 27. We have deleted “obviously” and changed the sentence in the methods section: “The pulse waves of the invasive recordings were visually checked and in that case cleared if they differed greatly from the mean (e. g. artifacts). As an example please see Fig 2a, where over 90 s, a total of two pulse waves had to be excluded. All other pulse waves were included in the analysis of central BP, what means that in the end several extrasystoles still were included.”

Moreover we deleted “(...)including criteria for high accuracy devices” in the abstract.

Additionally in the methods section we deleted “as precisely as possible” and added another sentence to clarify how zeroing was performed: “At the beginning of the invasive procedure, zeroing was performed precisely, having in mind that inaccuracy in zeroing will cause severe BP aberration. For calibration of each transducer the zero reference level for pressure measurement was set at midchest height, which was also used for balancing. Both calibration and balancing were checked before each measurement was performed.”

Abstract, line 25: “Invasive cBP recordings were prospectively compared to simultaneous non-invasive cBP estimations using the Antares algorithm, integrated into an oscillometric BP monitor.” Please explain the use of the term “prospectively” here.

Response 5: We deleted “prospective” and “prospectively” throughout the manuscript. Please see also response 2.

Abstract, line 28: “fulfilling the highest validation criteria”. Please be explicit about which criteria.

Response 6: Thank you very much for giving us the opportunity to make it very clear what is meant with “fulfilling the highest validation criteria”. This expression means that no matter which validation protocol you want to compare with, in each case Antares is within the highest validation criteria. We added to the Discussion section: An explanation of “best acceptable range” is given as follows: ““Best acceptable range” means for the 2017 ARTERY standard that the device tested passes the validation criteria with mean difference < 5 mmHg, SD<8 mmHg without fail criteria, for 2017 AAMI protocol that mean difference is below 5 mmHg with SD below 8 mmHg, for 2018 AAMI/ESH/ISO protocol if its estimated probability of a tolerable error (≤10 mmHg) is at least 85% and for British Hypertension Society (BHS) it passes grade A criteria. Consequently, for all named protocols, the results of the Antares algorithm to estimate cBP are within the best acceptable range.” We would be grateful if you could provide us with further validation protocols to further compare the quality of the algorithm.

Abstract, line 29: “Bland-Altman plot reveals good limits of agreement.” Please rephrase and quantify.

Response 7: We followed your advice and deleted this sentence from the abstract. The quantifications of the Bland-Altman plots are detailed in the legends of the Bland-Altman plots.

Abstract, lines 29-31: “Antares is the first algorithm for estimation of cBP that entirely fulfills the 2017 ARTERY and AAMI/ESH/ISO validation protocols including criteria for high accuracy devices.” See major comment #1 – the guidelines mentioned are for *device* validation. The present study shows that Antares works well *in a particular device*, which cannot be generalized to other devices.

Response 8: Thank you very much for stating that “Antares works well”. This is the main result of our validation study. Thank you for helping us to deliver this message in a high quality format. For this specific point please see response 1.

Table 1: Min and Max only represent outliers. Please report median, 25th and 75th percentile values, or mean and standard deviation.

Response 9: We integrated the standard deviation in table 1.

Table 2, first line: “% of goal”: this is only correct for the number of patients, for the other lines it represents % of patients

Response 10: Thank you very much. Correction is done.

Lines 106-110 refer to protocol differences, but the protocol has not been explained here yet. Please move down.

Response 11: Thanks again for precise scrutiny of the manuscript. We moved the sentence down to the paragraph where invasive central BP measurement is described.

Line 111: radial catheters: were these in the same arm that was used for brachial BP estimation? If so, does this influence the brachial BP measurement?

Response 12: We added to the Methods section: “In 99% of the cases invasive radial access was on the right site and the cuff for non-invasive recordings was placed on the left arm.”

Line 126/127: I would suggest excluding the two patients in whom the right arm was used.

Response 13: As stated in response 12, in 99% of the cases invasive radial access was on the right site and the cuff for non-invasive recordings was placed on the left arm. In 2 patients it was the other way round. That means that in no case the invasive measurement was performed on the same arm that was used for brachial BP estimation. Thus, we hope that with this additional information you can accept every patient included.

Lines 162-164: “zeroing was performed as precisely as possible”. What does that mean? How was the transducer height chosen?

Response 14: We changed the Methods section accordingly: “For calibration of each transducer the zero reference level for pressure measurement was set at midchest height, which was also used for balancing. Both calibration and balancing were checked before each measurement was performed.”

Line 176: delete “obviously”

Response 15: We have deleted “obviously” and changed the sentence in the methods section: “The pulse waves of the invasive recordings were visually checked and in that case cleared if they differed greatly from the mean (e. g. artifacts). As an example please see Fig 2a, where over 90 s, a total of two pulse waves had to be excluded. All other pulse waves were included in the analysis of central BP, what means that in the end several extrasystoles still were included.”

All figures: Please use English. Numbers, lines and labels should be much larger for readability.

Response 16: We have adapted all figures accordingly.

Figure 1: please align x-axes of the two panels, and use the same x-axis for panel 2a

Response 17: We changed the legend of figure 1b to make it clearer that the pulse waves displayed are the ones who are recorded during the deflation of the cuff. If we would adapt the x-axes of the two panels, then in figure 1b a big lag would be present. We hope that you can follow our changed presentation/legends of the figures. Please see response 18.

Figure 1b: multiple lines are plotted. What do they represent? Please use different colors/line types (e.g. dashing). Please also include an extra panel that shows one or two beats zoomed in, so we can appreciate the differences.

Response 18: We entirely agree with your comment and are thankful to give us the chance to correct this mistake. We have deleted Figure 1b and replaced it with a much clearer figure without unnecessary information like ECG. Accordingly, we have changed the legend of figure 1b and changed to English:

Figure 1b. Extracted oscillometric pulse waves of the same 65-year old lady during deflation of the cuff. The following signals are displayed: cuff pressure (cuff), extracted oscillometrical pulse waves (osci), and foot points for each identified pulse wave (foot) over time in seconds (s).

Figure 2a. this is nice, but what’s relevant is the beats that were recorded simultaneous with the brachial beats.

Response 19: Thank you very much for stating that the figure is nice. With the figures 2a and 2b we intend to clarify the invasive evaluation process. This seems to have a real meaning, see responses to reviewer 1. If you would prefer to delete figures 2a and 2b, please state again or find a solution together with reviewer 1 and/or the editor. Sure, we do, what you recommend. But we do not have access to pulse waves that may be displayed or calculated with Antares except for figures 1a and 1b. If you suggest removing these figures, please state again. We do have the calculated central pressures. And the present study is a validation study of cBP. Please see also response 20.

Accordingly we added funding information as follows: “Funding: The University Medicine Greifswald has received an unrestricted financial support by Redwave Medical GmbH for carrying out the study. The main parts of the study were financed by own funds of the participating centers. Redwave Medical GmbH did not have any influence on design and conduction of the study as well as on data analyses and writing of the manuscript, except providing detailed information about the algorithm requested by the reviewer and figures 1a and 1b.”

Additionally we added to the Acknowledgments: “We are grateful to Chris Stockmann (Redwave Medical GmbH) for providing information of the Antares algorithm and figures 1a and 1b.”

Please provide a figure where brachial and central blood pressure waves are plotted together in the same panel, so that waveform differences can be appreciated.

Response 20: We do not have access to the pulse waves of identical brachial and central pulse waves. We do have the invasive pulse waves as shown in figures 2a and 2b. I may state again that this is a validation study of cBP, not a paper of the manufacturer or anything else. Please see also responses 19 and 20.

Figures 4-9: please combine into one figure.

Response 21: We performed the combination of figures 4-9 into one figure. Then we had the strong feeling that the figures are a big chaos. So we went back one step and now have the figures again individually. If you want to see the figures combined into one figure, please drop a message and we believe it is convincing to see that it doesn’t work.

Figure 4/6/8 legend: “Dashed line: 45° reference”. Please replace with “Dashed line: line of identity”.

Response 22: Thank you very much. Figures 4, 6 and 8, legend: “Dashed line: 45° reference” was replaced by “Dashed line: line of identity”

Figures 5/7/9: please display x-axis numbers below (not inside) plot.

Response 23: Thank you very much again. Figures 5/7/9: x-axis numbers are now displayed below (not inside) plot.

Line 255 “the oscillometric blood pressure device”. Please replace with “the Custo Med Custo Screen 400 device”.

Response 24: we added information that it is the custo screen 400 monitor. We assume that you are talking about line 257 instead of 255, because in line 255 there is no BP device mentioned. If you want us not to change line 257 (first sentence of the discussion section) but something else, please state again.

Line 259 Please delete “prospective”.

Response 25: Done. Please see also responses 2 and 5.

Line 286: “Moreover, the data are pooled from three different study sites what could bear another potential error.” Please compute Bland-Altman statistics for these studies separately and compare the results (can be in a supplemental material).

Response 26: In an early phase of the invasive validation study we performed this analysis to get a feeling if there is a study site that might have integrated a systemic error or any other abnormalities. At this early time point of the study there was no difference in the performance of the different study sites. If you wish, we could do this analysis again and put into supplemental material.

Line 287 “within the best acceptable range”. Please reformulate.

Response 27: Thank you very much for giving us the opportunity to make it very clear what is meant with “within the best acceptable range”. This expression means that no matter which validation protocol you want to compare with, in each case Antares is within the best acceptable range. We added to the Discussion section: An explanation of “best acceptable range” is given as follows: ““Best acceptable range” means for the 2017 ARTERY standard that the device tested passes the validation criteria with mean difference < 5 mmHg, SD<8 mmHg without fail criteria, for 2017 AAMI protocol that mean difference is below 5 mmHg with SD below 8 mmHg, for 2018 AAMI/ESH/ISO protocol if its estimated probability of a tolerable error (≤10 mmHg) is at least 85% and for British Hypertension Society (BHS) it passes grade A criteria. Consequently, for all named protocols, the results of the Antares algorithm to estimate cBP are within the best acceptable range.” We would be grateful if you could provide us with further validation protocols to further compare the quality of the algorithm. We strongly believe that if we do not use the short term of e.G. “within the best acceptable range”, then it will be too much repetition to state all the protocols Antares is fulfilling with “grade A criteria” and others.

If you want us to repeat again which protocol criteria are fulfilled, sure, we will do that.

Lines 290-297: I suggest moving these to the limitations section.

Response 28: We moved the entire paragraph to the limitations section.

Line 300: “lack” -> “lag”

Response 29: Thank you very much for excellent scrutiny of our manuscript. Now in the Limitations section: “Lack” is corrected to “lag”.

Lines 300-301: “There was no lack of time between those measurements. However, a 100% synchronization via R-peak was not done.” These two sentences contradict each other. Why wasn’t an R-wave synchronization performed?

Response 30: Please see response 3. Actually during the conceptualization process of the study we were discussing this point intensively. We came to the conclusion that we might follow the 2013 AAMI protocol, what is also accepted following the 2017 ARTERY statement (see below). AAMI has included criteria that should guarantee hemodynamic stability. We do have invasive measurements of 90 s length in two of the three study sites. This gives us the chance to define the hemodynamic stability of the patient quite well and come close to the “true” central BP of this particular patient. The longer the period of invasive recording, the closer one gets to the “true” central BP. From a clinical point of view we are interested in “true” central pressure. In a clinical setting beat-to-beat-synchronization would be non-sense. And we really believe in clinical demands. Thus, we decided to compare the non-invasive acquisition of the oscillometric pulse waves with the longest possible period of invasive recordings to come close to the “true” central pressure.  In our opinion another advantage in following the AAMI protocol is that if other groups perform invasive measurements and compare their results with Antares, their results should be closer to our results even if they do not have exact simultaneous measurements. We may have better correlations in the present study if we analyze the precisely simultaneous period, but we may be farer away from the “true” cBP of the patient. Additionally we have understood the 2017 ARTERY statement in that way that this would be acceptable. The statement claims that the “non-invasive central BP values must be compared with the intra-arterial central BP (reference) values averaged over a time-period matching the deflation cycle of the non-invasive device and recorded under stable conditions, ideally simultaneously, or as contemporaneous as possible.” We definitely fulfill this criterion with having the non-invasive measurement covered by the invasive recordings in parallel. The ARTERY statement even would allow a comparison of invasive and non-invasive measurements with some additional requirements. Thus, during conceptualization process, we decided to go that way, we believe that this is reliable and even do hope more that you may can accept our thoughts.

In the limitations section we have described that the oscillometric recordings are 100% within the invasive recordings (examination order: invasive recording started, oscillometric recording started, oscillometric recording finished, invasive recording finished) and that we did not do a beat-to-beat-synchronization. Please allow us to give further explanation if required to the already given information in the limitation section. We added the limitations section:

“The invasive and non-invasive measurements were performed at the end of each cardiac catheter simultaneously. There was no lag of time between those measurements. However, 100% synchronization via R-peak was not done. To the best of our knowledge this should be acceptable because following the 2013 AAMI protocol, physiologic pressure fluctuations already were taken into account and should be covered by the length of the invasive measurements (20s in Greifswald, 90s in Bad Oeynhausen and Bad Berka, respectively). From a clinical point of view it may be advantageous to compare the estimated cBP with the invasive measurement procedure that comes as close as possible to the “true” central pressure. To average the cBP of an invasive recording of 90s might be closer to the “true” cBP than a shorter period of 20 or 30 seconds. The study should evaluate robustness and reliability for use of Antares in real life scenarios, where R-peak synchronization is not done. Therefore we followed the 2013 AAMI protocol and used the concept of hemodynamic stability of an entire invasive measurement, which timewise envelops the oscillometric measurement.”

Line 326 “blood pressure device”: please add “blood pressure device studied here”

Response 31: We added as requested “studied here”.

Lines 339-342: No funding information is provided. Please provide funding information.

Response 32: We added funding information as follows: “Funding: The University Medicine Greifswald has received an unrestricted financial support by Redwave Medical GmbH for carrying out the study. The main parts of the study were financed by own funds of the participating centers. Redwave Medical GmbH did not have any influence on design and conduction of the study as well as on data analyses and writing of the manuscript, except providing detailed information about the algorithm requested by the reviewer and figures 1a and 1b.”

Additionally we added to the Acknowledgments: “We are grateful to Chris Stockmann (Redwave Medical GmbH) for providing information of the Antares algorithm and figures 1a and 1b.”

Reviewer 3 Report

As stated, the study follows and passes the antares 2017 critera, which seem reasonable. The exception is for central diastolic blood pressure where there are too few individuals at the high pressure to decide whether the estimate or accurate or not. The abstract correctly avoids claiming to measure cDBP well. I am not familiar with ANSI/AAMI/ISO 81060-2:2013. The study seems altogether sensible.

minor issue: an 'and' in the author list problbly means a comma.

Author Response

Point 1: As stated, the study follows and passes the Artery 2017 criteria, which seem reasonable. The exception is for central diastolic blood pressure where there are too few individuals at the high pressure to decide whether the estimate or accurate or not. The abstract correctly avoids claiming to measure cDBP well. I am not familiar with ANSI/AAMI/ISO 81060-2:2013. The study seems altogether sensible.

Response 1: Sure, you are absolutely right in stating that cDBP at extreme pressures faces too few individuals. This is why we have dedicated a whole paragraph to this limitation in the Limitations section. Please let us know if we should even strengthen or expand this limitation. The status quo is:

”The AAMI protocols as well the ARTERY protocol suggest that an indicative range for invasive central systolic BP may be below/equal 100 mmHg in 5 % of the readings and above/equal 160 mmHg in 5 % of the readings and invasive central diastolic BP may be below/equal 60 mmHg in 5 % of the readings and above/equal 85 mmHg in 5 % of the readings. In the validation protocols this BP criterion is categorized as “may” what means that this criterion provides further guidance and is not a “must” or “should” recommendation. Though we have measured invasively in total 191 patients and could use data of 145 patients for the analysis, we could include in the extreme BP ranges only three patients for central SBP below/equal 100 mmHg and 17 patients for central DBP of above/equal 85 mmHg. The corresponding Bland-Altman plots have a good level of accordance without any significant trend. However, it has to be stated that further invasive comparisons have to show if within these extreme BP ranges Antares works similarly robust as shown for the other ranges.”

Point 2: minor issue: an 'and' in the author list probably means a comma.

Response 2: I agree with you but I have to refer to the Journal. They edited the manuscript with the result of the “and”. I am not too much familiar with the editing requirements of JCM. Thus, I have to refer and ask JCM if this is okay or if it has to be changed?

Thanks a lot for your Review.

Round 2

Reviewer 1 Report

I thank the authors for comprehensively addressing my comments. I have a few minor comments that should be addressed.

Comment 1. The first response, the section related to the algorithm does not appear to have been added to the methods as stated.

Comment 2: Line 128: typo: 'motoring' should read 'monitoring

Comment 3: The length of the introduction could be reduced by removing some of the extra detail on the AAMI/ISO guidelines.

Author Response

Comments and Suggestions for Authors

I thank the authors for comprehensively addressing my comments. I have a few minor comments that should be addressed.

Comment 1. The first response, the section related to the algorithm does not appear to have been added to the methods as stated.

Response 1: We deeply apologize for this negligence! Yes, you are "damned" right; I am very sorry (JB, my fault). In the Methods section, a sentence has now been dropped to avoid too many repetitions, and the specified algorithm details are now integrated (methods section, lines 155-162).

Comment 2: Line 128: typo: 'motoring' should read 'monitoring

Response 2: Done. Thanks for excellent scrutiny of the manuscript!

Comment 3: The length of the introduction could be reduced by removing some of the extra detail on the AAMI/ISO guidelines.

Response 3: Some details regarding the AAMI/ISO protocols were deleted from the introduction (lines 61, 63, 65).

Thanks again for significant and excellent contribution to enhance the quality of the manuscript!

Reviewer 2 Report

Thank you for all your elaborate responses. Please find my response to some of your remarks below.

I’d like to further discuss my major comments #1 and #3. To #1 you replied:

Response 1: Thank you for giving us the opportunity to further explain the principles standing behind the Antares algorithm. We appreciate very much to follow your demand to comment on the points raised by you above. We do it the best we can. But to some extend we have to refer to Redwave Medical because they developed the algorithm and may provide you with more information in detail. Sure, they provided us, the authors and the ones who performed the validation study with helpful information about the Antares algorithm before we started the present study. We are very much willing to offer this information. Actually you state that “This implies that the algorithm has to do two things: 1) properly describe the actual pressure amplification (like a type-I device should), but also 2) correct for the difference between an invasive and non-invasive (cuff) blood pressure measurement. In the case of Antares, both are integrated into one algorithm.” Additionally you state “Because Antares includes *both* steps as described above”. We hesitate to answer because you are the esteemed reviewer, but this is not true. Antares turns the custo screen 400 BP monitor very clearly into a type II device and not into a type I device. This is stated many times in the manuscript and described in the methods section in more detail. The process of recalibration is a step of pure preprocessing that is not displayed and not included in any report. This is an invasive validation of the central BP and not of other variables that might be calculated by Antares. If Antares is doing that, this would demand a separate validation study.

Thank you for your reply, sorry for some confusion that may have arisen. I fully agree with your statement “Antares turns the custo screen 400 BP monitor very clearly into a type II device.” This is exactly the conclusion of your study. What I tried to point out is that Antares itself is not a device, but an algorithm. The presented study validates the custo screen 400 BP monitor as a central blood pressure monitor – and that works well. However, this does not mean that if Antares is put into another device, that this device will work equally well. We simply cannot conclude that from this study.

The 2017 ARTERY statement defines type I and type II as follows: ”Type I—device purports to give an estimate of central BP relative to measured brachial BP (i.e. relatively accurate pressure difference between central and peripheral sites). Type II—device purports to estimate the intra-arterial central BP (i.e. relatively accurate absolute central BP value despite inaccuracy at the peripheral site).” This is why we have chosen to validate against the invasive intra-arterial pressure and not to the invasive pulse pressure amplification as it would have been necessary for an invasive validation of a type I device.  We are sorry but we do not see at all how Antares will be an algorithm for turning a BP monitor into a type I device. Thus, the deduction you stated might be subject of rethinking.

Thanks for pointing out the 2017 ARTERY statement again, which strengthens my point: Antares is not a DEVICE. These guidelines pertain to DEVICES.

For type 1, this is less of a problem: you can simply feed the device algorithm a peripheral blood pressure wave, and then see whether the central bp wave makes sense (you don’t have to worry about the cuff-vs-invasive difference).

Antares (or some preprocessing step therein) also corrects for (or incorporates in another way) this cuff-vs-invasive difference. This difference is much more device-specific than the (physiological) difference between peripheral and central BP.

To be clear: Antares turns the custo screen 400 monitor into a type 2 device (not type 1) – we agree on this.

To the best of our knowledge, Antares was developed to be integrated in several regular blood pressure monitors. The custo screen 400 monitor is a regular 24h-ABPM monitor. When the Antares algorithm is integrated, then it can provide central blood pressures that are passing criteria from 2017 ARTERY statement, 2013 AAMI protocol, 2018 AAMI/ESH/ISO criteria and BHS grade A criteria (see responses 3, 4, 6 and 27), each with the highest defined criteria. Antares is a novel type of algorithm which has been designed device independent.

That’s good, but I’m saying that the raw signals from a BP monitor ARE device-dependent.

Therefore, on one hand it requires access to the unfiltered raw data of any blood pressure cuff. On the other hand, it does not follow the operational steps of current devices of first handling pressure amplification and afterwards correcting the difference between invasive and oscillometric measurements (or vice versa). Consequently, since Antares is not performing those two steps, it does not need to be tailored for different devices.

Think about this reasoning. Why would the correction between invasive and oscillometric would be so hard to do and be device-dependent, and then magically Antares would work in each device…?

Nevertheless, it is clearly mentioned, that if Antares can be integrated reliably with similar robust results in other oscillometric devices has to be proven.  Please see limitations section: “If Antares can be integrated reliably with similar robust results in other oscillometric devices has to be proven.”

Thank you for this statement, which is exactly what I mean.

An excellent example of an invasive validation of an algorithm to estimate central BP is done by Dr. Thomas Weber (Weber T, Wassertheurer S, Rammer M, Maurer E, Hametner B, Mayer CC, Kropf J, Eber B. Validation of a brachial cuff-based method for estimating central systolic blood pressure. Hypertension 2011;58:825–832.). Together with the inventor of the ARCSolver he validated this algorithm that basically is doing the same like Antares.

No, the ARCSolver transforms a brachial PRESSURE WAVEFORM into a central PRESSURE WAVEFORM. It doesn’t address the cuff-versus-invasive issue.

The main message is the same within the present paper, that ARCSolver is invasively validated, not just the oscillometric device. Even if ARCSolver runs in SphygmoCor and analyzes its pressure waves, no one would argue that this is a validation of the SphygmoCor device and not the ARCSolver algorithm.

But Antares doesn’t take a pressure wave as an input. That is the key difference. Antares takes a potentially highly device- and cuff-specific oscillogram and turns that into a central BP.

So according to the literature the present validation and interpretation seems to be in line with former publications of algorithm validations.

Maybe things aren’t as bad as I present them here, but we don’t know, and simple need those studies done. We simply cannot conclude from the present study that Antares works well for all blood pressure monitors.

However, to address your criticism even more, we added in the abstract “The Antares algorithm turns the custo screen 400 BP monitor into a type II device.”

Thank you for this addition. Taken together, can you please also change/add these two related things:

Abstract Lines 21-22: “The present study is an invasive validation of the Antares algorithm”: Please add “in the custo screen 400 BP monitor”.

Conclusion, line­ 371: please add to the start of the sentence: “When implemented into the device, Antares fulfills…”

To major comment #3 you replied:

Response 3: Actually during the conceptualization process of the study we were discussing this point intensively. We came to the conclusion that we might follow the 2013 AAMI protocol, what is also accepted following the 2017 ARTERY statement (see below). AAMI has included criteria that should guarantee hemodynamic stability.

But why guarantee hemodynamic stability only for the invasive and not for the noninvasive measurement? If you just use the same beats, you don’t need any stability!

We do have invasive measurements of 90 s length in two of the three study sites. This gives us the chance to define the hemodynamic stability of the patient quite well and come close to the “true” central BP of this particular patient. The longer the period of invasive recording, the closer one gets to the “true” central BP. From a clinical point of view we are interested in “true” central pressure. In a clinical setting beat-to-beat-synchronization would be non-sense.

We (reviewer #1 also raised this issue) are not suggesting anything like performing beat-to-beat synchronization when this device is used normally; we are only suggesting to do this for this study, i.e., to compare apples to apples. I understand that beat-to-beat synchronization can be tricky, but at least compare two signals of equal length. Can you (at least for a subset) perform this analysis, and show that this didn’t influence your findings?

And we really believe in clinical demands.

This comment does not add anything to this discussion.

Thus, we decided to compare the non-invasive acquisition of the oscillometric pulse waves with the longest possible period of invasive recordings to come close to the “true” central pressure.  In our opinion another advantage in following the AAMI protocol is that if other groups perform invasive measurements and compare their results with Antares, their results should be closer to our results even if they do not have exact simultaneous measurements. We may have better correlations in the present study if we analyze the precisely simultaneous period, but we may be farer away from the “true” cBP of the patient.

Or, you may be introducing a systematic error due to comparing periods of different length. You may be right that things get even better when the exact periods are compared, but maybe they don’t… Again, please perform this analysis (at least for a subset of subjects; can be in a supplement). It will only strengthen the paper further if it works better on a beat-to-beat (or same timeframe) basis.

Additionally we have understood the 2017 ARTERY statement in that way that this would be acceptable. The statement claims that the “non-invasive central BP values must be compared with the intra-arterial central BP (reference) values averaged over a time-period matching the deflation cycle of the non-invasive device and recorded under stable conditions, ideally simultaneously, or as contemporaneous as possible.” We definitely fulfill this criterion with having the non-invasive measurement covered by the invasive recordings in parallel. The ARTERY statement even would allow a comparison of invasive and non-invasive measurements with some additional requirements. Thus, during conceptualization process, we decided to go that way, we believe that this is reliable and even do hope more that you may can accept our thoughts.

In the limitations section we have described that the oscillometric recordings are 100% within the invasive recordings (examination order: invasive recording started, oscillometric recording started, oscillometric recording finished, invasive recording finished) and that we did not do a beat-to-beat-synchronization. Please allow us to give further explanation if required to the already given information in the limitation section. We added the limitations section:

“The invasive and non-invasive measurements were performed at the end of each cardiac catheter simultaneously. There was no lag of time between those measurements. However, 100% synchronization via R-peak was not done. To the best of our knowledge this should be acceptable because following the 2013 AAMI protocol, physiologic pressure fluctuations already were taken into account and should be covered by the length of the invasive measurements (20s in Greifswald, 90s in Bad Oeynhausen and Bad Berka, respectively). From a clinical point of view it may be advantageous to compare the estimated cBP with the invasive measurement procedure that comes as close as possible to the “true” central pressure. To average the cBP of an invasive recording of 90s might be closer to the “true” cBP than a shorter period of 20 or 30 seconds.

The study should evaluate robustness and reliability for use of Antares in real life scenarios, where R-peak synchronization is not done.

I would personally delete this preceding sentence about R-peak synchronization. This “real life” reasoning is simply incorrect – nobody is suggesting to do R-peak synchronization during normal use.

Therefore we followed the 2013 AAMI protocol and used the concept of hemodynamic stability of an entire invasive measurement, which timewise envelops the oscillometric measurement.”

I’d like to further discuss two of my minor comments:

Comment/response 16: Thanks for updating the figures. However, text in figures 1a, 1b, 2a, 2b is still way too small. Figures 3-9 are good – please compare and note the difference.

Comment/response 24: Sorry for my typo.

Thanks for all your replies!
